# The Effects of Acute and Chronic Aerobic Activity on the Signaling Pathway of the Inflammasome NLRP3 Complex in Young Men

**DOI:** 10.3390/medicina55040105

**Published:** 2019-04-15

**Authors:** Iman Khakroo Abkenar, Farhad Rahmani-nia, Giovanni Lombardi

**Affiliations:** 1Department of Exercise Physiology, Faculty of Sport Sciences, University of Guilan, Rasht 4199613776, Iran; frahmani2001@yahoo.com; 2IRCCS Istituto Ortopedico Galeazzi, Laboratory of Experimental Biochemistry and Molecular Biology, via Riccardo Galeazzi 4, 20161 Milano, Italia; giovanni.lombardi@grupposandonato.it; 3Department of Physiology and Pharmacology, Gdańsk University of Physical Education and Sport, Kazimierza Górskiego 1, 80-336 Gdańsk, Pomorskie, Polska

**Keywords:** acute and chronic aerobic exercise, inflammation, inflammasome complex

## Abstract

*Background and Objectives*: The results of the studies show that the intensity and volume of aerobic exercise activity produce different responses of the immune system. This study aims to show how the signaling pathway of the inflammatory NLRP3 complex is influenced by the acute and chronic effects of moderate and high-intensity aerobic exercises in young men. *Materials and Methods*: Accordingly, 60 healthy (BMI = 23.56 ± 2.67) young (24.4 ± 0.4) students volunteered to participate in the study that was randomly divided into two experimental (*n* = 20) groups and one control (*n* = 20) group. The training protocol started with two intensity levels of 50% for a moderate group and 70% of maximum heart rate for high group for 30 min and then continued until reaching 70% (moderate group) and 90% (high group) of the maximum heart rate, respectively. Using Real Time-PCR method, the expression of NLRP3 gene and ELISA- were measured by IL-1β, IL-18. *Results*: The results showed that acute aerobic exercise with moderate intensity had no significant effect on the expression of NLRP3 gene and serum levels of IL-1β and IL-18 cytokines (*p* > 0.05) when acute exercise, with high intensity, begins an initiation of the activity of the inflammatory complex with elevated serum levels of IL-1β, IL-18, and NLRP3 gene expression (*p* < 0.05). In addition, chronic exercise with moderate intensity significantly reduced the expression of NLRP3 gene and serum levels of IL-1β, IL-18 cytokines (*p* < 0.05). In the case of chronic exercise with high intensity, a significant increase in expression of gene, NLRP3 and serum levels of IL-1β, IL-18 cytokines were observed (*p* < 0.05). *Conclusions*: Generally, it can be concluded that chronic exercise with moderate intensity is effective in decreasing the expression of the inflammasome and inflammation.

## 1. Introduction

It has been shown that exercise is beneficial for our health; however, it affects the immune function both positively and negatively [1,2,3]. Two divisions that have different functions build the immune system: The innate immunity, known as the first line of defense, and the adaptive (or humoral) immunity, which upon activation sets up a particular reaction and immunological memory to each infectious agent. Studies have reported that diseases are preventable through the advantageous effect of well-ordered moderate-to-vigorous physical activity [4]. It has also been shown that based on the intensity, type, and duration, local and systematic inflammation is initiated by the exercises that contribute to the release of both pro- and anti-inflammatory cytokines [5]. The exercise-induced inflammatory response is highly defined so that, in general terms, during physical activity, the pro-inflammatory mediators prevail while during the recovery period the anti-inflammatory response takes place [6].

The first stage of the inflammatory response to pathogens or stressful stimuli begins with the activation of specific receptors called (pattern recognition receptors—PPRs) that are expressed by virtually all the eukaryotic organisms [7]. In response to infections, the main defense cells, including macrophages, monocytes, dendritic cells, neutrophils, and epithelial cells express PRRs. According to the recent studies, another set of PPRs (NOD-like receptors, NLRs) are probably involved in identifying the so-called pathogen-associated molecular patterns (PAMPs) and damage-associated molecular patterns (DAMPs) and, hence, trigger the inflammatory response. Unlike the membrane-associated Toll-like receptors (TLRs), NLRs are cytoplasmic receptors that are actively involved in identifying the microbial components or high-risk signals. NLR family members play a role in the assembly of molecular platforms, the inflammasome, which are activated by cellular infection or stress that help the pro-inflammatory cytokines such as interleukin-1β (IL-1β) and IL-18 to mature to participate in the innate immune defense mechanisms. Besides the involvement in cytokine maturation, inflammasome play a role in a highly inflammatory form of cell death called pyro ptosis [8]. NLRs are a very large family divided into four subfamilies. The NLRP subfamily is the one most involved in the development of the inflammasome complex, and the NLRP3 is the most prominent member acting as an indicator of inflammasome activation [9]. IL-1β, a key downstream effector of inflammasome activation, affects both local and systemic inflammation significantly. Note that the expression and secretion of IL-1β should be managed accurately at the transcriptional and post-translational levels. IL-1β, along with TNFα, participates in innate immunity and inflammation. IL-1β also regulates inflammatory responses, but it is involved in the regeneration process by increasing the expression of the matrix metalloproteinase [10]. Another relevant inflammatory cytokine in the inflammasome signaling pathway is IL-18 that is activated and released by the cysteine-protease caspase-1. Thus, the processing and secretion of IL-18, as well as secretion of other proteins, such as IL-1β, and fibroblast growth factor-2 (FGF-2) are dependent on caspase 1 [11]. According to the clinical findings, there is an association between high levels of plasma IL-1β and IL-18 and several inflammatory diseases like gout, diabetes, and Alzheimer’s disease atherosclerosis.

Hence, by abating, the inflammatory status may have beneficial effects in preventing these conditions. Also, the inflammatory properties can be used to increase the immunogenicity of chemotherapy drugs and work as an adjuvant that can modulate the immune responses to antigens to improve them [12]. Hence, it is very important to know the levels of inflammatory response and inflammasome complex [13]. Sports researchers in the field of sports immunology are interested in studying the effects of short-term and long-term exercise with varying degrees of intensity on the immune system. The part of the results of the study by Mardena et al. (2016) to compare the chronic effect of aerobic training vs. resistance training on inflammasome activation and ceramide release in Wistar led to controversy. In comparison with the control group, the expression of IL-18 increased considerably in both the resistance and aerobic groups; however, contrary to the control group, IL-1β was reduced in both training groups. Increased expression of inflammasome components was observed in the aerobic group, whereas it was reduced in the resistance training group [14]. Mejinas Pena et al. conducted a study to investigate the effect of an eight week resistance training protocol in elderly men; they reported that the signaling pathway of the inflammasome was considerably reduced in the training group in comparison with the control group. In addition, the NLRP3 gene expression in PBMC was declined along with the serum level of caspase 1 [15]. Since the key pathogenic role of the inflammatory complexes and the positive effects of physical activity, studies investigating how exercising may turn off these mechanisms are desirable. In fact, the inflammasome is probably activated by the effects of the acute and chronic exercises but how exercise effect on inflammasome activation and these effects are unclear. Hence, this study aims to show how the signaling pathway of the inflammatory NLRP3 complex is influenced by the (moderate and high) acute and chronic aerobic activity in young men.

## 2. Methodology

### 2.1. Study Cohort

Sixty Sports Science undergraduate male students took part in this study voluntarily. Contributors had been recruited from the community and gave informed consent before participation. All subjects enjoyed good health and were physically fit; they were not receiving any medications and did not follow any diet regimen. Inclusion criteria were as follows: age 19 to 27 years, being able to run at least 3 km outside with assisting devices as required, being able to run at least 20 min on the treadmill at 8 km/Hr., having stable cardiovascular situation metabolic equivalents, and not presently collaborating informal rehabilitation. Exclusion criteria were as follows: decreasing aerobic endurance, and threats to physical health.

### 2.2. Training Protocol

The subjects were first recruited, then, they were put into two groups: the active (*n* = 40) and the control groups (*n* = 20). In the active group, the subjects were randomly assigned to either the moderate-intensity exercise group (*n* = 20) and the high-intensity exercise group (*n* = 20). The training protocol consisted of the acute sessions and chronic sessions of Nordic walking on a NordicTrack machine (Nordic Ski Machine, Logan, UT, USA). After a familiarization session with the exercise, the subjects were asked to do exercise at either moderate- or high-intensity depending on heart rate as monitored by the Monark Sports Watch (Monark Exercise, Stockholm, Sweden). Prior to the protocol and after the training, the subjects were measured in terms of height and weight. Using a caliper and implementing the Jackson–Pollock 3-point (abdomen, super iliac and triceps) method, body fat percent was determined indirectly (Laffayette, 01127 mod, USA) [16]. The exercise program started three days after the initial blood sampling. The program included a three month aerobic training (three days/week) in the active group, while the control group maintained the daily lifestyle and abstained from any physical training during the same period. At the beginning of each training session, the subjects performed a five min warm up exercise protocol. The exercise protocol, instead, was started at 50% HRmax for 30 min and then continued at 70% HRmax for the remaining 40 min, in the moderate-intensity group. In the high-intensity group, the exercise started at 70% HRmax for 30 min and it was continued at 90% HRmax for the remaining 40 min. During the test, the instructor encouraged subjects to keep the right intensity and the total length of the exercise was increased week from five min to 20 min. All training sessions were held on Sundays, Tuesdays, and Wednesdays at 16:00 [17]. After three days from the completion of the training protocol, blood samples were taken from the subjects. The procedures were according to the ethical standards of the committee for experimentation on humans and the Declaration of Helsinki from 1975 and its revised edition in 1983. The study was approved by the Local Institutional Ethics Committee of Guilan University (Ir.gums.rec 1396.45).

### 2.3. Blood Sample Preparation

To collect the samples, 20 mL of blood was taken from the left-hand antecubital vein under fasting state (12 h), 72 h before the start of the first exercise session, in EDTA-anticoagulated tubes. Blood was collected after 48 h of the first bout of exercise and at the end of the training (12 weeks). Peripheral-blood mononuclear cells (PBMCs) were divided from the fresh whole-blood using density gradient centrifugation on Ficoll separating solution (Biochrom, Berlin, Germany) [18]. For each sample, three 15 mL centrifuge tubes were used to layer 6 mL of 1:1 Hank’s balanced salt solution (HBSS)-diluted blood onto 4 mL of Ficoll. The suspension was put in a centrifuge for 30 min at 275 g at room temperature (RT). Through manual pipetting, the mononuclear cell layer was taken away and was washed once in Hank’s solution, and centrifuged for 10 min at RT and 450 g after the wash. The cells were washed two times in PBS, and suspended in 1 mL of PBS once more. 

### 2.4. Reverse Transcription and Quantitative Real-Time Polymerase Chain Reaction

A RiboPure-Blood Kit (Ambion Life Technology, Grand Island, NY, USA) was used according to the manufacturer’s instructions to isolate Total RNA from PBMC. For quantification, the fluorescent method Ribogreen RNA Quantitation Kit (Molecular Probes Life Technology, Grand Island, NY, USA) was implemented. RNA was incubated with DNase I RNase-free DNase (Ambion, Loughborough, UK) to remove the residual genomic DNA. A reverse transcriptase kit (Promega, Stockholm, Sweden) was applied to synthesize the first strand cDNA. The negative control (no transcriptase control) was performed in parallel. TaqMan Universal PCR Master Mix (Applied Biosystems, Waltham, MA, USA) was applied on an ABI 7000 (Applied Biosystems) following the manufacturer’s cycling parameters to amplify cDNA. The 2-ΔΔCTmethod was applied to specify the relative changes in gene expression levels. The cycle number showing the transcripts (CT) was normalized to the cycle number of β-actin, referred to as ΔCT. Table 1 shows the primer sequence and 𝛽-actin was used as an endogenous control.

### 2.5. Cytokine Assays

ELISA kits (MABTECH, Sweden) were used based on the manufacturer’s instructions to specify the concentrations of Supernatant Cytokines (IL-18, IL-1β). The absorbance was read by an ELISA reader (Awareness Technology, FL, USA). IL-1β, IL-18 levels were expressed as pg/mL. The lowest limits of detection were, 20 pg/mL for IL-1β, 9.8 pg/mL for IL-18; respectively.

### 2.6. Data Analysis

The data were expressed as mean ± SD. To determine the mean and standard deviation of descriptive variables, descriptive statistics were used. The Kolmogorov–Smirnov test was applied to test the normality of distribution. All data were found to be approximately normally distributed. To examine the homogeneous variances of variable distribution, Leven’s test was used. Data were analyzed for main effects using a one-way ANOVA for repeated measures. Statistical significance was accepted when *p* < 0.05. To assay the between and within -group differences at each time point, the post hoc LSD test was used. Data analyses were done with SPSS (IBM Version 16, Windows 7, New York, NY, USA) and Microsoft Excel 2007.

## 3. Results

The participants’ characteristics before the start of the protocol are presented in Table 2. All three groups were matched before starting the protocol, the results of one-way ANOVA test showed that there is not any significant difference between groups in all variables. Also, the PBMC NLRP3 expression and plasma IL-1β and IL-18 were not affected by the acute bout of moderate-intensity aerobic exercise. On the contrary, when the same exercise was performed at high intensity, the inflammasome was triggered which was marked by a significant increase of IL-1β and IL-18 plasma concentrations (*p* < 0.05), and expression of NLRP3 in PBMCs (*p* < 0.05). Nevertheless, the expression of NLRP3 in PBMCs and the plasma levels of IL-1β and IL-18 (*p* < 0.05) were decreased considerably following the moderate chronic 12 week aerobic exercise. Whereas, NLRP3 gene expression (*p* < 0.05). and plasma IL-1β and IL-18 were increased (*p* < 0.05) after the high-intensity training both NLRP3 gene expression and plasma IL-1β and IL-18 were increased (*p* < 0.05). In the same period, the analyzed parameters were unchanged in the control group. Results are summarized in Figure 1A–C.

## 4. Discussion

According to the findings of this study, the expression of NLRP3 in lymph-monocytes and plasma concentrations of IL-1β and IL-18 were not affected by the acute moderate-intensity aerobic exercise and, hence, it does not activate any signaling pathway triggering the inflammasome. On the contrary, when the same exercise is performed at high intensity, activation of the inflammasome complex is possible due to the significantly increased IL-1β, IL-18 plasma concentrations and the increased NLRP3 expression in inflammatory cells. In addition, our findings indicated that the expression of NLRP3 as well as the plasma, of IL-1β and 1L-18, was decreased following the moderate-intensity aerobic training, whereas, the 12 week high-intensity aerobic training exerted the reverse effects.

Consistent with our findings, previous studies by Gleeson et al. showed that effective, high-intensity aerobic exercises increased the TLR4 gene expression and increased serum levels, as well as protein expression of inflammatory cytokines such as C-reactive protein (CRP), TNF-α, IL-6 IL-10, IL-1β [17,19]. Only a few parallel studies have shown that moderate-intensity exercising is accompanied by a decrease in the TLR4 signaling pathway activation and inflammasome complex, or that it has no significant effect on that route, as we found in the current study [20,21].

Numerous studies have reported that the effects of exercise on health are dependent on the intensity and duration of exercise protocol [22,23]. It was shown that moderate physical activity (about 60–80% of maximal oxygen uptake) affects the function of various tissues positively and boosts the status of reduction-oxidation balance [24]; however, it is likely that an acute bout of heavy exercise could lead to several side effects including oxidative stress and inflammatory response [25]. It has been shown that the inflammatory signaling was considerably decreased in the hippocampus in ovariectomized mouse compared to the untrained control group following four weeks of resistance training. In the same work, it was shown that also the expression of NLRP3 gene was decreased together with the blood concentration of caspase-1 and IL-1β, IL-18 [26]. According to Lee et al., oxidative stress was increased in mitochondria following the high-intensity acute exercise accompanied by heightened activation of the inflammasome complexes pathway. This phenomenon was considered necessary to increase the defense response against pathogens [25]. 

Clinical and pathological studies have reported the role of the inflammasome complex in cellular and molecular immunology, which led to autophagy and a strong correlation with various diseases, like cancer, chronic inflammations, and type 2 diabetes [27,28]. As it was pointed out earlier, NLRP3 inflammasome, the well-characterized inflammasome, mediates between the expression and function of IL-1β and IL-18(29). Our findings show that inflammation was decreased in young men following moderate exercise as proved by the reduced level of pro-inflammatory cytokines IL-1β and IL-18 in PBMC. Furthermore, the results of the present study indicated that the expression of NLRP3 is directly decreased by exercise. 

Studies have shown that exercise may be effective in limiting inflammasome activation, but the role of exercise intensity remains unclear. It is not clear how NLRP3 inflammasome activation is suppressed by exercise [29,30]. It seems that chronic high-intensity aerobic training due to the role of cellular autophagy inflammasome, autophagy can be regulated by inflammasome components, which can escalate free radicals and increased the up-regulation of the inflammasome. Whereas chronic aerobic training with moderate intensity can increase the blood flow and oxygen uptake to peripheral tissues, thus, reducing the process of capillary leukemia and reducing the inflammatory cytokines that predispose to inflammasome [31]. Furthermore, acute moderate intensity practice has a beneficial effect on the immune system. This type of exercise is able to increase VO2max and down regulation of inflammatory cytokines [30].

It is probable that exercise training does not allow the deleterious immune cells to accumulate that contribute to inflammation and insulin resistance; it hampers the loss of adipose tissue immune cells that are involved significantly in the regulation of the inflammatory levels, therefore It can be an important lifestyle factor in reducing inflammation, inflammasome complex and the oxidative capacity of an organism [32]. Also, acute high-intensity practice stimulates and invokes inflammatory pathways and inflammasome. As it was noted earlier, inflammasome are localized in the cytosol and their activation could be triggered by exercise signaling when they are generated in this compartment; however, this could probably be an adjuvant effect on the immune system [33].

## 5. Conclusions

In conclusion, the results of the present study suggest that some of the inflammatory indices can be reduced following exercise training, particularly endurance-type training. In fact, in most of the diseases and inflammatory conditions, the inflammasome NLRP3 is up-regulated, and exercise with suitable intensity can decrease inflammasome. This was the first study to show how the signaling pathway of the inflammasome NLRP3 complex in young men is influenced by acute and chronic exercise with moderate and high intensities. Further studies are suggested to determine which intensity and protocols are better for improving the efficiency of the immune system. 

## Figures and Tables

**Figure 1 medicina-55-00105-f001:**
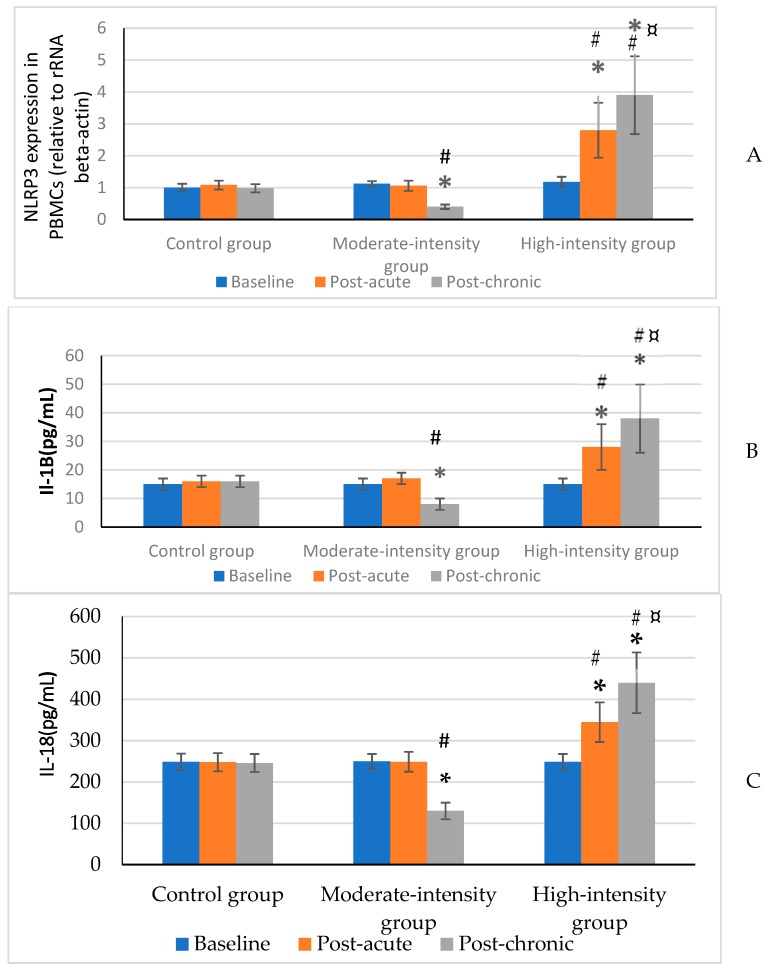
NLRP3 expression in PBMCs (relative to rRNA beta-actin) (**A**), and IL-1β (**B**) and IL-18 (**C**) plasma concentrations in controls and training subjects (moderate-intensity and high-intensity groups) at baseline (pre-test) after 48 h (post-acute) and 12 weeks (post-chronic) from the start of the protocol. Data are presented as mean ± SD. * significant difference within groups with baseline (*p* ≤ 0.05). # significant difference between groups with the control group (*p* ≤ 0.05). ¤ significant difference between moderate and high intensity groups (*p* ≤ 0.05).

**Table 1 medicina-55-00105-t001:** Primers sequences used in qPCR assays.

For: 5′-TGGACTTCGAGCAAGAGATG-3′ β-actin
Rev: 5′-GAAGGAAGGCTGGAAGAGTG-3′
For: 5′-ATGAAGATGGCAAGCACCCG-3′ NLRP3
Rev: 5′-CTACCAAGAAGGCTCAAAGACGAC-3′

**Table 2 medicina-55-00105-t002:** Subject characteristics.

	High-Intensity (*n* = 20)	Moderate-Intensity (*n* = 20)	Control (*n* = 20)	F	*p*
Age (years)	24.1 ± 1.8	23.65 ± 2.43	23.5 ± 3.21	3.09	0.06
Height (m)	1.73 ± 0.49	1.76 ± 1.01	1.74 ± 0.44	0.63	0.53
Weight (kg)	72.98 ± 1.71	73.45 ± 2.28	75.35 ± 2.13	1.64	0.20
BMI (kg/m^2^)	23.77 ± 2.11	23.84 ± 1.92	24.43 ± 3.66	0.58	0.55
Body fat (%)	19.22 ± 2.39	19.45 ± 2.55	20.62 ± 2.75	2.11	0.13

Data are reported as mean ± SD.

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
