# Peer review of "The Effects of Acute and Chronic Aerobic Activity on the Signaling Pathway of the Inflammasome NLRP3 Complex in Young Men"

_medicina, 2019, doi:10.3390/medicina55040105_

Reviewer 1 Report

The manuscript  entitled "The effects of acute and chronic aerobic activity on the signaling pathway of the inflammasome NLRP3  complex in young men" concludes that endurance chronic aerobic activity with moderate intensity is effective against inflammasome and inflammation. The findings are interesting to readers but:

1) English need extensive improvement. It is difficult to interpret abstract too.

2)  Authors should provide good study design. They should not just provide comparison of weight and BMI of control and training participants but should also compare moderate and high intensity group.

3) Authors should provide selection criteria for groups in detail.

3) Figure 1. Figure legend post acute and post chronic are not correct captions for control group.

4) Please show activation of inflammasome through some experiments: western blots for pro IL-1beta, IL-1 beta, pro and cleaved caspase. 

5) Please show PCR for caspase, ASC, IL-1 beta

Author Response

Reviewer#1

The manuscript entitled "The effects of acute and chronic aerobic activity on the signaling pathway of the inflammasome NLRP3 complex in young men" concludes that endurance chronic aerobic activity with moderate intensity is effective against inflammasome and inflammation. The findings are interesting to readers but:

1) English need extensive improvement. It is difficult to interpret abstract too.

The manuscript has been thoroughly revised by an English native speaker. A certification letter about editing is available

2) Authors should provide a good study design. They should not just provide a comparison of weight and BMI of control and training participants but should also compare the moderate and high-intensity group.

The authors agree with the reviewer about this suggestion. Table 1 has been implemented with additional information about the subjects

3) Authors should provide selection criteria for groups in detail.

                Inclusion and exclusion criteria have been entered in the text (page3, in the study cohort section)

3) Figure 1. Figure legend post-acute and post chronic are not correct captions for the control group.

                Figure legend has been amended

4) Please show activation of inflammasome through some experiments: western blots for pro-IL-1beta, IL-1 beta, pro, and cleaved caspase.

Since the low RNA yield from whole blood neither additional RNA nor protein were made available for further analyses. We are aware of the fact that the investigated cytokines are not exclusively expressed by blood cells and that, hence, their circulating levels are the net result of a systemic integration of a wide variety of signals. However, their circulating levels mirror the systemic inflammasome activation and, hence, they represent a useful and “diagnostic” index in this context.

5) Please show PCR for caspase, ASC, IL-1beta.

We have raw data for IL-1b, IL-18 by Elisa and expression for Nlrp3 by real-time _pcr.

Finally, thank you for taking the time to readout wanted to revise.

I hope you confirm them.

Waiting for your kind reply.

Yours sincerely

Reviewer 2 Report

In the manuscript entitled “The effects of acute and chronic aerobic  activity on the signaling pathway of the inflammasome  NLRP3 complex in young men”,  the authors examine the possible difference of the effects between acute and chronic aerobic activity on the inflammasome NLRP3 complex.  Using  Real Time-PCR method (measuring NLRP3 expression level) and ELISA (determining plasma concentrations of the IL-18 and IL-1β),    the authors clearly show that the levels of the target molecules from the acute moderate-intensity group are unchanged, while levels from the chronic moderate-intensity group are decreased. The authors also show that the levels from high-intensity groups both acute and chronic are increased. The authors conclude that both acute and chronic high-intensity exercises significantly increased IL-1β, IL-18 plasma concentrations and NLRP2 expression in inflammatory cells, while moderate-intensity training didn’t change the activation of the inflammasome complexes (acute) or reduced the activation (chronic).

Overall the study is clear in its outcome.  The findings from the study are of benefit to the exercise health community.

There are a few minor points that could be addressed to make the manuscript stronger.

Line 167: “…was marked by a significant increase of IL-1β and IL-18 plasma concentrations expression of NLRP3 in PBMCs (p< 168 0.05).” It might be needed to add  “, and ” between “concentrations” and “expression”.

Line 171: “…both NLRP3 gene expression and plasma IL-1β and IL-18 were increased (p< 0.05).” It is not clear to me what does “both” mean.

Line 184: “… it does not active any signaling pathway triggering the inflammasomes.”  From the Figure 1, it is clear that the acute moderate-intensity exercise has negative activations to the inflammasome complexes.  

Line 186: “…is possible …” It is not clear what do the authors try to say using “possible”.

Between line 198 and line 209: there is little confusions in the discussion; it might be better to discuss “moderate-intensity exercise” and “high-intensity exercise”  in separate paragraphs.

It may be useful to add one or two references to the discussion between line 218 and line 227.

Author Response

Reviewer #2

In the manuscript entitled “The effects of acute and chronic aerobic activity on the signaling pathway of the inflammasome NLRP3 complex in young men”, the authors examine the possible difference of the effects between acute and chronic aerobic activity on the inflammasome NLRP3 complex. Using Real Time-PCR method (measuring NLRP3 expression level) and ELISA (determining plasma concentrations of the IL-18 and IL-1β), the authors clearly show that the levels of the target molecules from the acute moderate-intensity group are unchanged, while levels from the chronic moderate-intensity group are decreased. The authors also show that the levels from high-intensity groups both acute and chronic are increased. The authors conclude that both acute and chronic high-intensity exercises significantly increased IL-1β, IL-18 plasma concentrations and NLRP2 expression in inflammatory cells, while moderate-intensity training didn’t change the activation of the inflammasome complexes (acute) or reduced the activation (chronic).

Overall the study is clear in its outcome. The findings from the study are of benefit to the exercise health community.

There are a few minor points that could be addressed to make the manuscript stronger.

Line 167: “…was marked by a significant increase of IL-1β and IL-18 plasma concentrations expression of NLRP3 in PBMCs (p< 168 0.05).” It might be needed to add “, and” between “concentrations” and “expression”.

                Amended

Line 171: “…both NLRP3 gene expression and plasma IL-1β and IL-18 were increased (p< 0.05).” It is not clear to me what does “both” mean.

                The sentence has been rewritten

Line 184: “… it does not active any signaling pathway triggering the inflammasomes.”  From Figure 1, it is clear that the acute moderate-intensity exercise has negative activations to the inflammasome complexes.

                Probably this referee was confused by the grammar. This new version would be clear

Line 186: “…is possible …” It is not clear what do the authors try to say using “possible”.

                Other factors can be involved in this phenomenon (e.g., oxidative stress).

Between line 198 and line 209: there is little confusions in the discussion; it might be better to discuss “moderate-intensity exercise” and “high-intensity exercise” in separate paragraphs.

The sentence has been rewritten and now it clear.

It may be useful to add one or two references to the discussion between line 218 and line 227.

                Two more references have been added

Finally, thank you for taking the time to readout wanted to revise.

I hope you confirm them.

Waiting for your kind reply.

Your sincerely

Round  2

Reviewer 1 Report

Though as a reviewer I think that authors should have had done few more experiments as suggested in first round of review but owing to limited samples as mentioned by authors the manuscript can be accepted in present form. 

Author Response

hello and regard

we re edited article(second time) by  english expert native .and we sent highlight  file and certification letter for medicina editor assistant.
